# Estimating the cumulative incidence of SARS-CoV-2 with imperfect serological tests: Exploiting cutoff-free approaches

**Judith A. Bouman**[1]*, **Julien Riou**[2], **Sebastian Bonhoeffer**[1], **Roland R. Regoes**[1]*

**1** Institute of Integrative Biology, ETH Zurich, Zurich, Switzerland, **2** Institute of Social and Preventive Medicine (ISPM), University of Bern, Bern, Switzerland

* judith.bouman@env.ethz.ch (JB); roland.regoes@env.ethz.ch (RRR)

## Abstract

Large-scale serological testing in the population is essential to determine the true extent of the current SARS-CoV-2 pandemic. Serological tests measure antibody responses against pathogens and use predefined cutoff levels that dichotomize the quantitative test measures into sero-positives and negatives and use this as a proxy for past infection. With the imperfect assays that are currently available to test for past SARS-CoV-2 infection, the fraction of sero-positive individuals in serosurveys is a biased estimator of the cumulative incidence and is usually corrected to account for the sensitivity and specificity. Here we use an inference method—referred to as *mixture-model approach*—for the estimation of the cumulative incidence that does not require to define cutoffs by integrating the quantitative test measures directly into the statistical inference procedure. We confirm that the mixture model outperforms the methods based on cutoffs, leading to less bias and error in estimates of the cumulative incidence. We illustrate how the mixture model can be used to optimize the design of serosurveys with imperfect serological tests. We also provide guidance on the number of control and case sera that are required to quantify the test's ambiguity sufficiently to enable the reliable estimation of the cumulative incidence. Lastly, we show how this approach can be used to estimate the cumulative incidence of classes of infections with an unknown distribution of quantitative test measures. This is a very promising application of the mixture-model approach that could identify the elusive fraction of asymptomatic SARS-CoV-2 infections. An R-package implementing the inference methods used in this paper is provided. Our study advocates using serological tests without cutoffs, especially if they are used to determine parameters characterizing populations rather than individuals. This approach circumvents some of the shortcomings of cutoff-based methods at exactly the low cumulative incidence levels and test accuracies that we are currently facing in SARS-CoV-2 serosurveys.

## Author summary

As other pathogens, SARS-CoV-2 elicits antibody responses in infected people that can be detected in their blood serum as early as a week after the infection until long after

**Data Availability Statement:** All R-code files are available from the gitlab database here: https://gitlab.ethz.ch/jbouman/pist.

**Funding:** RRR gratefully acknowledges funding from the ETH Zurich and the Botnar Research Centre for Child Health (grant number 2020-FS-354). The funders had no role in study design, data collection and analysis, decision to publish, or preparation of the manuscript.

**Competing interests:** The authors have declared that no competing interests exist.

recovery. The presence of SARS-CoV-2 specific antibodies can therefore be used as a marker of past infection, and the prevalence of seropositive people, i.e. people with specific antibodies, is a key measure to determine the extent of the SARS-CoV-2 pandemic. The serological tests, however, are usually not perfect, yielding false positive and false negative results. Here we exploit an approach that refrains from classifying people as seropositive or negative, but rather compares the antibody level of an individual to that of confirmed cases and controls. This approach leads to more reliable estimates of cumulative incidence, especially for the low prevalence and low test accuracies that we face during the current SARS-CoV-2 pandemic. We also show how this approach can be extended to infer the presence of specific types of cases that have not been used for validating the test, such as people that underwent a mild or asymptomatic infection.

This is a *PLOS Computational Biology* Methods paper.

## Introduction

Past SARS-CoV-2 infection can be detected with serological tests, which classify individuals as sero-positive when their antibody level exceeds a predefined cutoff-value. Such serological tests are imperfect; currently available tests have high specificity but relatively low sensitivity [1–7]. Thus, they can give rise to false positive and false negative results. Such incorrect test outcomes are particularly problematic when the serological tests are used to determine the immune status of individuals. To estimate the cumulative incidence in a population, however, the accurate determination of the serological status of each individual is not required, as it is possible to correct for the sensitivity and specificity of the serological test.

It is well known that the proportion of positive tests in a serosurvey is a biased estimator of cumulative incidence. Tests with a high specificity and low sensitivity, for example, underestimate the cumulative incidence because the low sensitivity leads to a high frequency of false negatives. Post-hoc corrections of the binomial estimator have been developed and refined over the past decades [8–10], and have been applied to SARS-CoV-2 serosurveys [11]. More specifically, Rogan and Gladen describe a post-hoc correction that essentially subtracts the expected number of false positives and adds the expected number of false negatives to the binomial estimator [8]. However, this Rogan-Gladen estimator works well only for high cumulative incidence and relatively high test specificity [8]. Additionally, the uncertainty in the sensitivity and specificity is not taken into account by the Rogan-Gladen estimator of the cumulative incidence.

An estimator that considers the uncertainty in the sensitivity and specificity is, in this study, referred to as *the Bayesian framework*. In this framework, the uncertainty in the sensitivity and specificity is taken into account by using the number and test outcomes of control and case sera used for test validation in the estimation of the cumulative incidence [12–16]. (Basically, sensitivity and specificity, as well as their uncertainty, are estimated as the Binomial probabilities of detecting true positives and negatives, respectively.) Similar to the Rogan-Gladen estimator, this framework dichotomizes the data using a predefined cutoff by assigning a positive or negative test result for each individual in the cohort. Thus the Bayesian framework is also cutoff-based.

In this study, we use an alternative approach to estimate cumulative incidence that does not rely on a predefined cutoff. This approach has been developed to study the cumulative incidence of infections with various pathogens and is referred to as *mixture model* [17–23].

Throughout this study, we will also refer to this method as *mixture model method*. Rather than relying on dichotomized serological test data and correcting for the test's sensitivity and specificity, this method uses the quantitative test measures together with data on the distribution of these measures in controls and confirmed cases. Serological tests differ in the quantitative test measures they yield. Some measure optical densities, other neutralization titers (NT50). The mixture-model approach can be applied to all these measures irrespective of the type of assay that is used, as long as the measures of the serosurvey and test validation data have the same units. If there is significant variation in measures between laboratories, as suggested by Bendavid et al. [11], measurements should be standardized or performed in the same laboratory.

Using simulated serosurveys, we investigated the statistical properties of mixture-model estimators of cumulative incidence, and compare them to cutoff-based estimators. Further, we conducted power analyses to determine the required size of a serosurvey for serological tests with varying accuracies. We also studied how the statistical power of the mixture model estimator is affected by the number of control and case sera used for the validation of the serological assay. This sheds light on the under-explored relationship between the effort put into test validation on the one hand, and the serosurvey on the other.

Last, we addressed an issue that can only be studied with the mixture model approach: the detection of a discrepancy between the serological observations in the epidemiological surveys and the quantitative test measures collected during the validation phase of the serological assay. There is evidence that individuals who underwent an asymptomatic or mild infection have lower antibody levels than those who had a severe infection [4, 24, 25]. A past asymptomatic or mild infection is therefore detected with a lower sensitivity than a severe infection for most serological assays [4, 24]. This results in a biased estimate of the cumulative incidence with the cutoff-based methods if the serological assay is not validated with a representative sample of cases. The mixture model approach, however, can detect such a discrepancy, correct for it, and even estimate the relative incidence of the category of samples that was excluded from the validation data. Thus, the mixture-model approach has the potential to map the diversity of disease profiles, and could shed light on the yet-to-be-determined frequency of asymptomatic infections with SARS-CoV-2.

## Results

### Mixture-model estimator outperforms cutoff-based methods

To compare the mixture model to the cutoff-based methods, we simulated serosurveys conducted with serological tests of varying accuracies. As a proxy for the accuracy, we selected the area under the ROC-curve (AUC-ROC) and varied it from 0.8 to 1. This range is consistent with the AUC-ROC values of most currently available SARS-CoV-2 antibody tests [4, 7, 26]. (The sensitivity and specificity corresponding to the standard cutoffs across the range of AUC-ROC values that we consider here, are given in S1 Fig) An overview of the cutoff-based methods can be found in Table 1.

In the simulated serosurveys, we assumed cumulative incidences of 1%, 4% and 8% and enrolled 10,000 virtual individuals and used 5,000 control and case sera for test validation. We then derived estimates of the cumulative incidence with the cutoff-based methods and the mixture-model approach, and repeated this 50 times for each test accuracy. We calculated 95% confidence intervals for the uncorrected and Rogan-Gladen corrected cutoff-based methods and the mixture model using a bootstrap procedure. A function for calculating these intervals is provided in the R-package. The uncertainty in the estimates from the Bayesian framework are determined based on the 95% credible interval.

**Table 1. Overview of methods for the estimation of cumulative incidence.**

| Name in this study | Cutoff-based | Used cutoff | Method of incorporating sensitivity and specificity | Estimated parameters | Ref |
|---|---|---|---|---|---|
| Uncorrected cutoff-based methods | yes | High specificity | None | Cumulative incidence | |
| | yes | Max Youden | None | Cumulative incidence | |
| Rogan-Gladen corrected cutoff-based methods | yes | High specificity | Post-hoc Rogan-Gladen correction | Cumulative incidence | [8] |
| | yes | Max Youden | Post-hoc Rogan-Gladen correction | Cumulative incidence | |
| Cutoff-based methods in a Bayesian framework | yes | High specificity | Simultaneous, Bayesian estimation of cumulative incidence, sensitivity and specificity | Cumulative incidence, sensitivity and specificity | [12, 14–16] |
| | yes | Max Youden | Simultaneous, Bayesian estimation of cumulative incidence, sensitivity and specificity | Cumulative incidence, sensitivity and specificity | |
| Mixture model | no | - | Mixture model for the estimation of cumulative incidence | Cumulative incidence | [17–23] |

S2 Fig illustrates the well-known fact that the observed fraction of positive serological tests is a biased estimator of the cumulative incidence. For both cutoffs we consider, the maximum Youden and the high specificity cutoffs, the bias is substantial, increases for test of lower accuracy, and, depending on the cutoff, can lead to over- of underestimation of the cumulative incidence.

The results of the analyses that take the sensitivity and specificity of the serological test into account are shown in Fig 1. We find that the Rogan-Gladen correction [8] (see Methods) results in unbiased estimations of the cumulative incidence (see Fig 1A). Note that for low cumulative incidence levels or low test accuracy, the corrected point estimate of the prevalence can become negative. This is due to the fact that the number of observed seropositives—being a realization of a stochastic process—can be smaller than the expected number of false positives. For the same reason, the Rogan-Gladen correction gives rise to more variable estimates compared to the mixture model for low cumulative incidence levels and low test accuracies (see Fig 1B). In contrast, the mixture-model estimator does not inflate the variation of point-estimates and does not result in negative estimates for any true cumulative incidence level or test accuracy.

The Bayesian framework results in slightly overestimated cumulative incidences for low incidence levels and low test accuracy. This is likely due to the flat priors that have been used. Fig 1B shows that the 95% credible intervals obtained by the Bayesian framework are larger than the 95% confidence intervals of the mixture-model estimator. Thus, the mixture model also outperforms the Bayesian framework.

## Mixture model leads to more reliable estimates of the temporal trends in cumulative incidence

In the ongoing SARS-CoV-2 pandemic, the cumulative incidence is still increasing, and an estimation of its temporal trend is an urgent public health objective. To assess how well the different methods can estimate temporal trends in cumulative incidence, we simulated serosurveys during an ongoing epidemic. In particular, we assumed that the cumulative incidence increases from 1.5% to 15%.

We found that the uncorrected cutoff-based methods both underestimate the temporal trend in the cumulative incidence (Fig 2). The Rogan-Gladen estimator removes the bias in the estimate, but introduces a large variation in the point-estimates of the temporal trend. This is similar for the cutoff-based methods within the Bayesian framework. The estimate for the

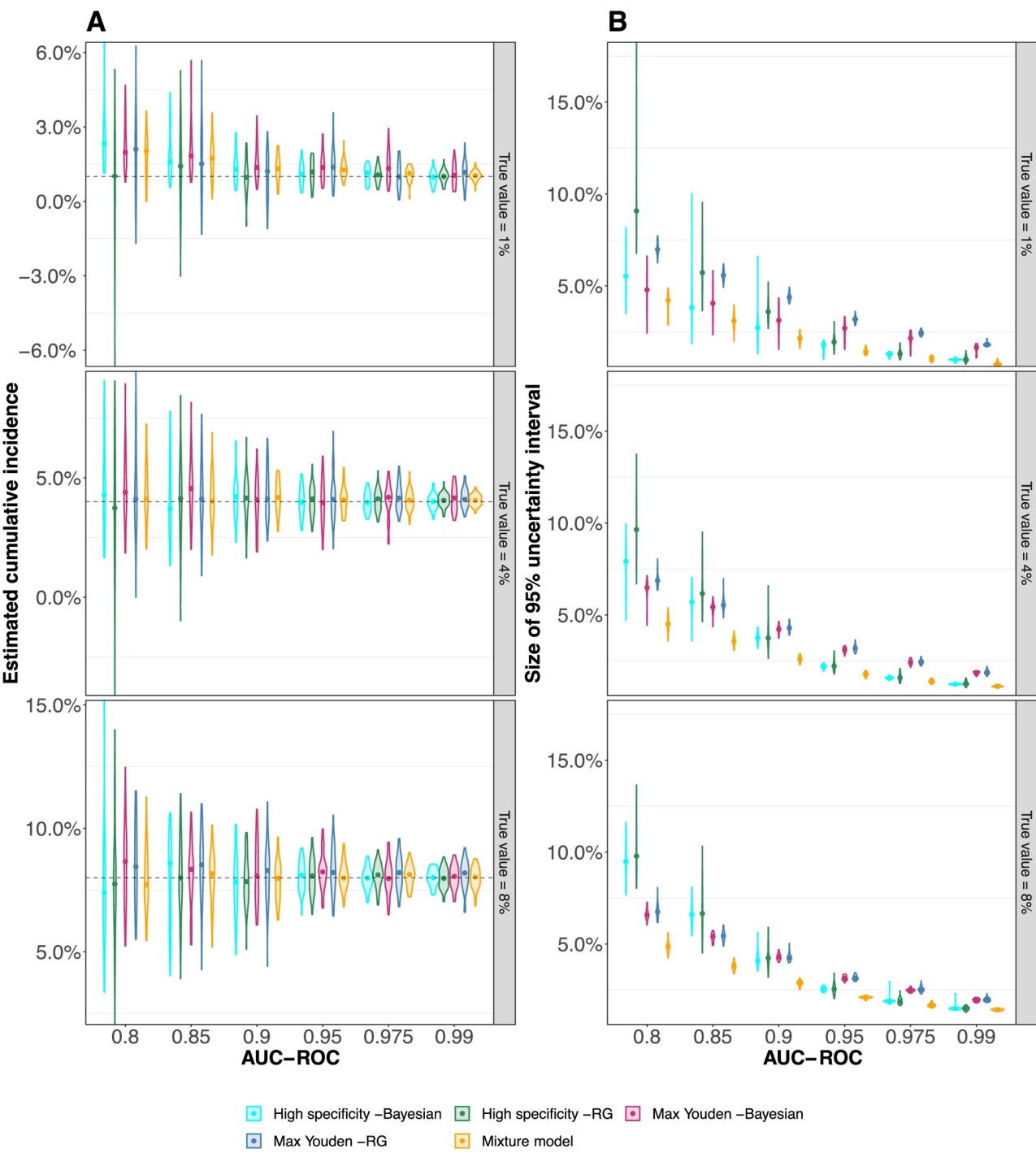

**Fig 1. Point-estimates of cumulative incidence using the cutoff-based methods and the mixture model.** Each violin represents 50 *in silico* serosurveys conducted with cohorts of 10, 000 virtual individuals and the points represent the median values. (A) Point-estimates of the cumulative incidence. The dashed line indicates the true cumulative incidence we assumed in the simulations. Please note that the scale of the y-axis differs between the sub-figures. (B) Size of the 95% uncertainty intervals. For the Rogan-Gladen and mixture-model estimates, the uncertainty intervals are the 95% confidence intervals, which we calculated with the bootstrap method (see Methods). For the Bayesian estimators, the uncertainty interval are the 95% credible intervals.

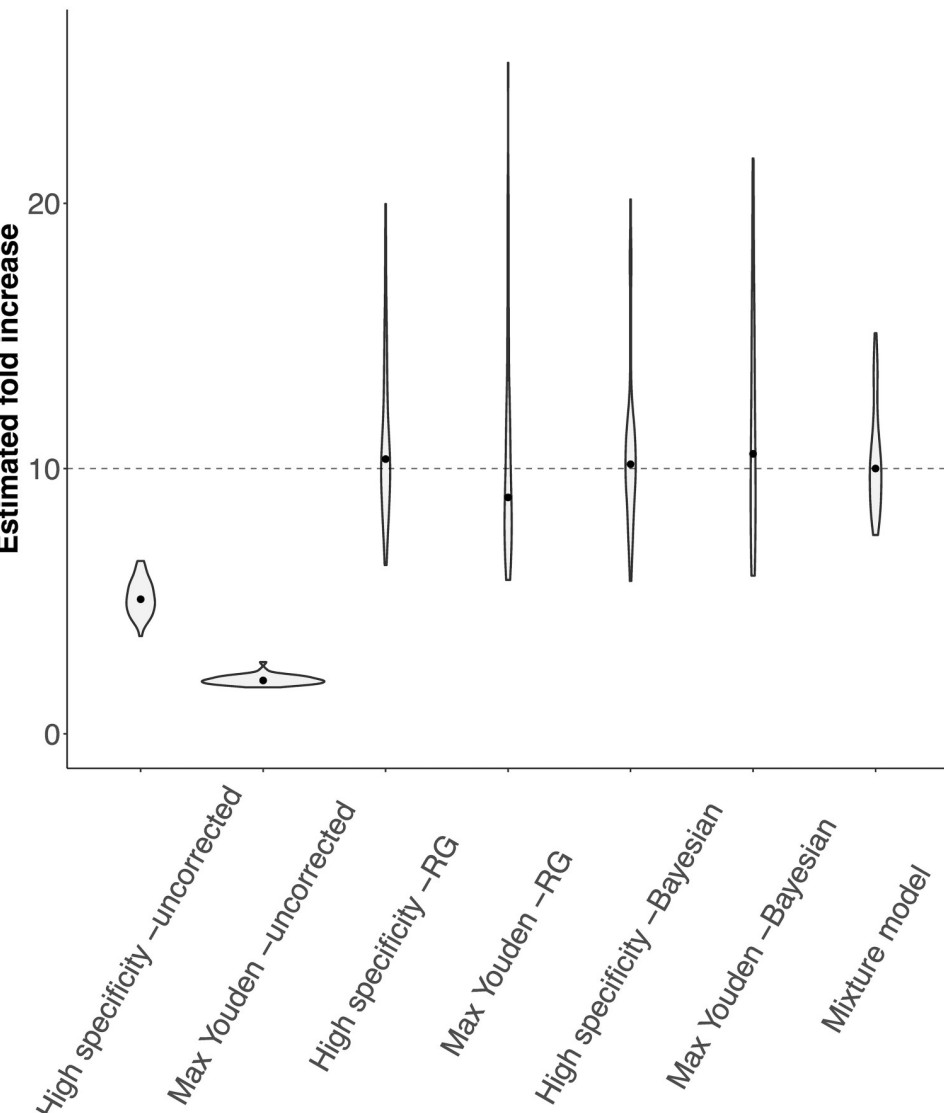

**Fig 2. Estimated fold increases in cumulative incidence for the cutoff-based methods and the mixture model.** In the simulated serosurveys, we assumed the cumulative incidence to increase from 1.5% to 15%, resulting in a true fold increase of 10 (dashed line). The violins show the distribution of 50 *in silico* serosurveys for both cumulative incidence levels conducted with cohorts of 10, 000 individuals and a test with an AUC-ROC value of 0.975. The dots indicate the median value.

temporal trend can even be negative for the reason mentioned above. In contrast, the mixture model leads to more reliable point-estimates of the temporal trend without inflating their variation.

## Low specificity/sensitivity can be compensated by enrolling more participants into the serosurvey

To understand how the test accuracy and the number of individuals enrolled in the serosurvey affect the statistical power of serosurveys analyzed with the mixture-model approach, we simulated serosurveys with tests characterized by varying AUC-ROC values and different numbers of individuals enrolled, and evaluated their success. A serosurvey was defined to be successful

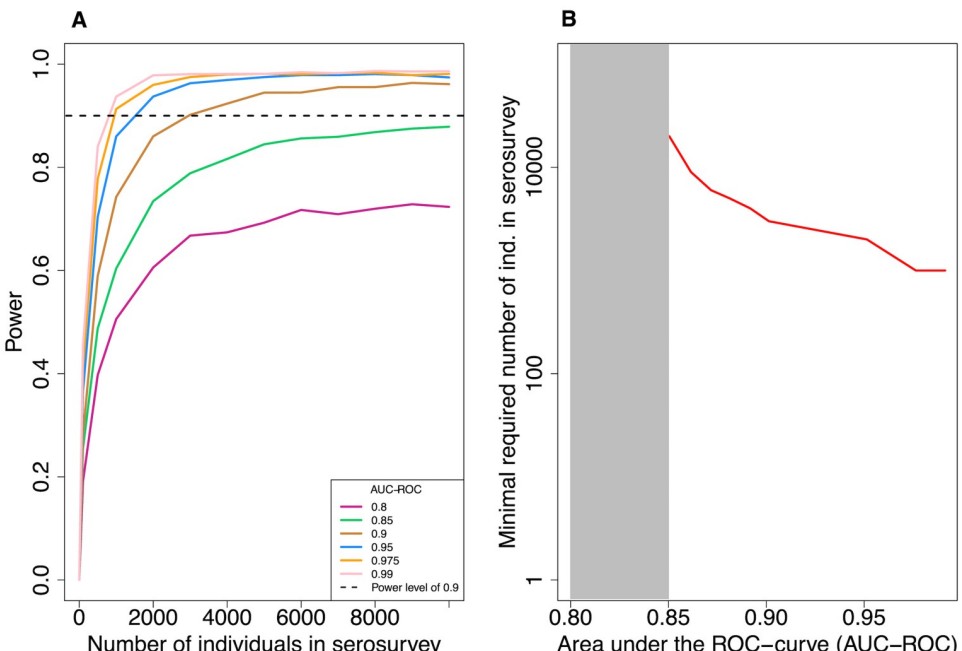

**Fig 3. Statistical power of the mixture model.** In all simulations, the number of control and case data is fixed to 5, 000 each and the true cumulative incidence level is 8%. (A) Statistical power versus the number of individuals in the serosurvey for varying levels of test accuracy (AUC-ROC). The power is calculated as the fraction of simulated serosurveys that resulted in a cumulative incidence estimate that is within 25% of the true cumulative incidence and for which the true cumulative incidence level lies within 2 standard deviations of the estimated value. Each point in the graph represents the result of 3, 000 *in silico* serosurveys. (B) The minimal number of virtual individuals necessary to obtain a statistical power of 0.9 over a range of AUC-ROC values.

if the estimate of the cumulative incidence was sufficiently close to the true cumulative incidence (see Methods). The statistical power is defined as the fraction of successful *in silico* serosurveys.

We find that a lower accuracy of the serological test can be compensated by higher sample sizes (Fig 3A), but only if the accuracy is not too low. For example, to achieve a statistical power of 0.9, 1, 000 individuals need to be enrolled into the serosurvey for a high-accuracy test (AUC-ROC = 0.975). In contrast, for a lower accuracy test (AUC-ROC = 0.9) 3, 000 are required (Fig 3B). However, for a test accuracy lower than 0.9 enrolling even up to 10, 000 individuals does not increase the power of the test above 0.9.

## The number of control and case sera used for validation of serological tests impacts the reliability of the cumulative incidence estimate

The mixture-model estimator relies directly on the quantitative test measures and their distribution for control and case sera. Therefore, the reliability of the method is expected to depend on the precision with which these distributions have been determined, which depends on the number of control and case sera used in test validation. Here, we asses how the number of control and case sera influences the variation of the estimated cumulative incidences and the power of the serosurvey (see Methods).

In our simulations, the confidence interval of the estimated cumulative incidence decreases with the number of control and case sera used for the validation of the test (Fig 4C). As a result, the statistical power increases for higher numbers of control and case sera (Fig 4D). Additionally, we find that, with increasing test accuracy, the number of control and case sera required

to obtain a statistical power of 0.9 decreases. For a true prevalence of 8% and 10, 000 individuals in the serosurvey, as used in Fig 4D, and a AUC-ROC value of 0.9, 2, 500 control and case sera samples are needed to obtain a statistical power of 0.9 (Fig 4D). However, for a AUC-ROC value of 0.975, sampling 750 cases and controls would be sufficient.

A low number of control and case sera for the validation of the test can be compensated by a higher number of individuals in the serosurvey (Fig 4E). The relation between the number of control and case sera and the minimal number of individuals in the serosurvey depends on the accuracy of the serological assay.

### Detecting discrepancies between test validation and serosurvey data

The reliance of the mixture model on quantitative test data, allows us to investigate inconsistencies between the populations used for validating the test on the one hand, and the population recruited into the serosurvey on the other. For example, assume that the serological test has been validated only with sera from individuals who underwent a severe infection, in addition to sera from controls (see Fig 5A). If these severe cases used in the validation set are a good representation of the infections in the population, we would expect our serosurvey data to look like Fig 5C. However, if other types of infections are very prevalent, such as mild or asymptomatic infections, that are characterized by distributions of antibody levels very different from severe infections (see Fig 5B), the resulting serosurvey data would look like those shown in Fig 5D. We have extended the mixture model such that it can detect from the serosurvey data whether there exists an additional distribution of cases that was not included in the validation data.

We set up a simulation assuming three distributions of test measures. These distributions can be thought of as corresponding to severe and asymptomatic cases and controls, but could also describe classes of infections that differ by a factor other than disease severity. We further assume that the total cumulative incidence in the population is 8%, of which 20% were asymptomatic infections, consistent with the current estimate [27]. We then simulate a serosurvey with 10, 000 individuals that results in quantitative test measures randomly drawn from these three distribution according to their cumulative incidences.

First, we show that ignoring a missing class of case sera leads to an underestimation bias in the estimated cumulative incidence by analysing the simulated serosurvey data assuming a mixture model based on quantitative test measure of only severe cases and controls (see Fig 6A). This misspecified mixture model, however, is valuable as a null model against which a more complex likelihood can be tested. Fig 6A compares the misspecifation scenario to the scenario where both asymptomatic and severe infections have been included into the distribution of the case sera of the validation data.

Secondly, we fit the three-distribution likelihood to estimate the cumulative incidences of asymptomatic and severe cases, and the mean of the test measure distribution of asymptomatic cases. This analysis—without knowledge of the third distribution—succeeds in estimating the total cumulative incidence without bias (Fig 6B). It also yields estimates for the cumulative incidences for asymptomatic and severe cases separately (Fig 6B). The identifiability of the individual cumulative incidences depends strongly on how distinct the test measure distributions of asymptomatic and severe cases and the controls are. In practice, one can test for the existence of a missing case distribution by applying a likelihood ratio test on the extended model and the standard mixture model.

### Discussion

In this study, we evaluate the performance of a mixture-model approach in estimating the cumulative incidence of a population. Throughout this paper, we refrained from referring to

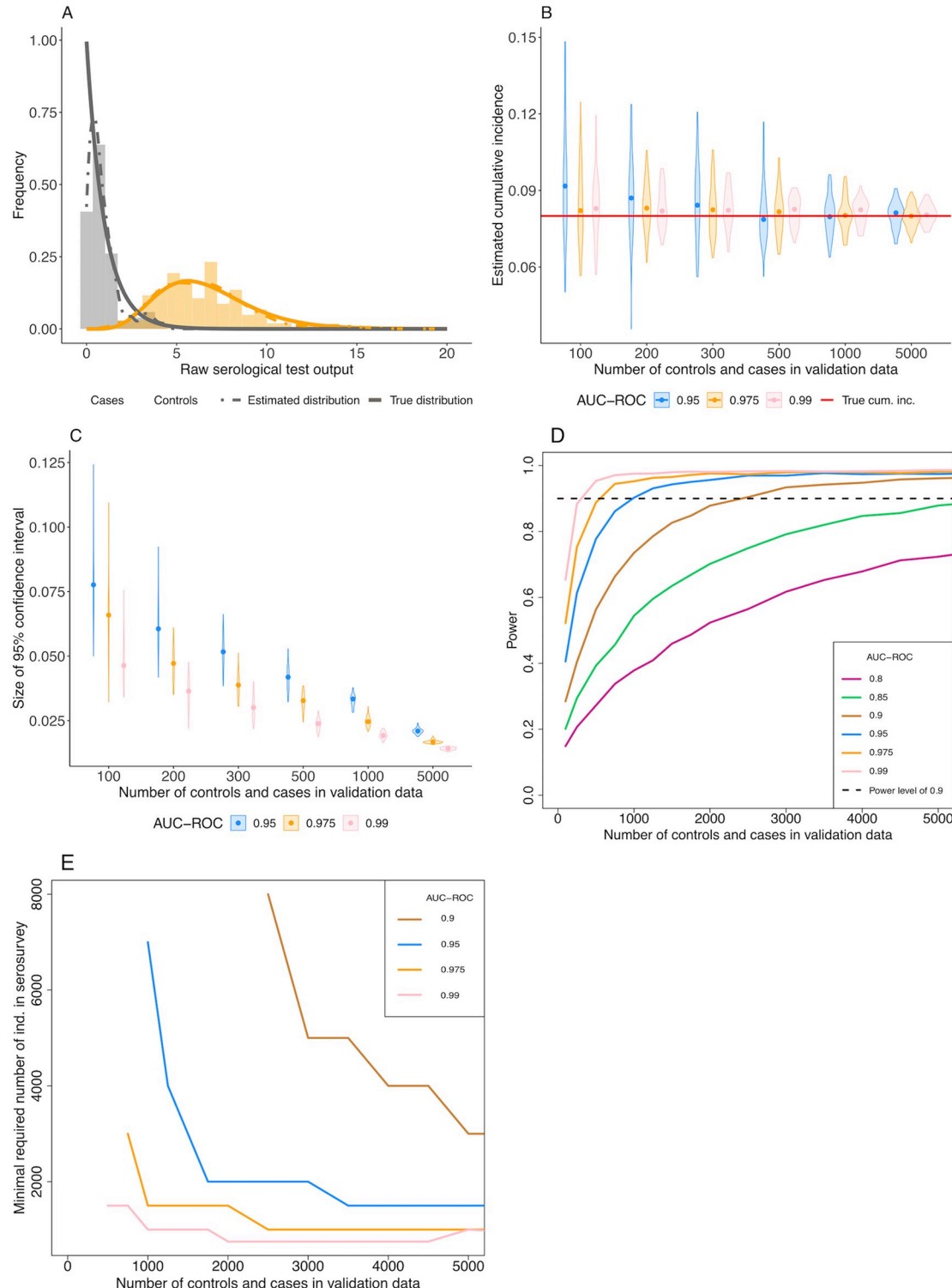

**Fig 4. Effect of varying the number of control and case sera used to calibrate the serological test.** (A) An example of the true distribution (solid lines) of the control (grey) and case (orange) sera, the data simulated from those distributions (histograms) and the inferred densities (dashed lines) used in the inference of the cumulative incidence. Here, 150 control and case sera have been simulated and the AUC-ROC value of the test is equal to 0.975. (B) Point estimates of cumulative incidence for various numbers of control and case sera used to calibrate the serological test and three AUC-ROC values. Each violin shows the distribution of the estimated cumulative incidence of 50 *in silico* serosurveys conducted with cohorts of 10, 000 virtual individuals. The red line shows

the true cumulative incidence we assumed in the simulated serosurveys (8%). (C) Size of the 95% confidence intervals of the estimated cumulative incidences. (D) Statistical power versus the number of control and case sera used in the validation data for varying levels of test accuracy (AUC-ROC). Each point in the graph represents the result of 3, 000 *in silico* serosurveys. (E) The minimal number of virtual individuals necessary to obtain a statistical power of 0.9 over a range of number of control and case sera in the validation data.

sero-prevalence for two reasons. First, sero-prevalence is the frequency of sero-positive individuals in a population and hence inherently coupled to cutoffs. Focusing on the distribution of antibody levels without dichotomizing into sero-positives and -negatives makes it more challenging to determine sero-prevalence. Second, serological tests are usually validated against pre-pandemic sera and sera from individuals with confirmed infection, and not against sera that do or do not contain pathogen-specific antibodies. As a consequence of this choice of positive and negative controls, false positives are individuals that have specific antibodies despite not having been infected, and not individuals for which the test wrongly shows they harbor specific antibodies while they actually do not. Correcting for false positives and negatives by both cutoff-based and mixture model methods therefore results in an estimate of cumulative incidence rather than of sero-prevalence. This distinction between sero-prevalence and cumulative incidence is common in epidemiological studies (see for example a metastudy on the cumulative incidence of the 2009 influenza pandemic by Van Kerkhove et al [28]).

The mixture-model approach uses the quantitative serological test measurements before they have been dichotomized, i.e. categorized as "positive" or "negative" with a cutoff. We confirm that cutoff-based estimators that do not correct for the sensitivity and specificity of the serological test can lead to strong biases in the estimate of cumulative incidence and its time trends, especially for situations of low cumulative incidence [8]. Even though the commonly

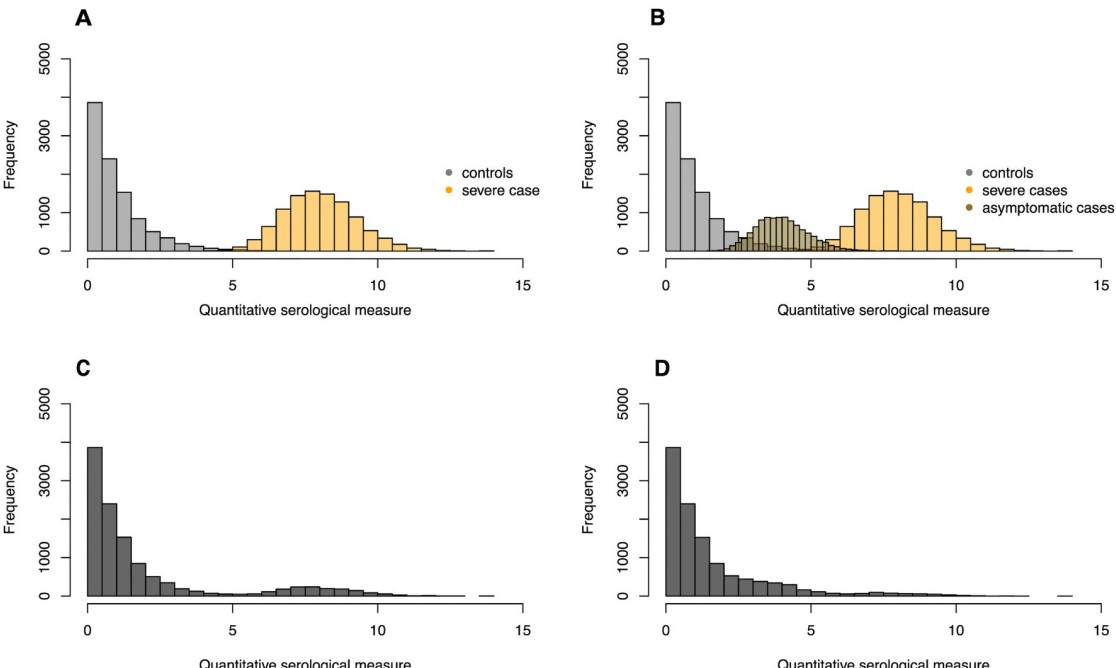

**Fig 5. Conceptual figure on how a discrepancy between the test validation and serosurvey data can be detected.** (A) Histograms of simulated validation data from controls and severe cases. (B) Histograms of simulated validation data from controls and severe and asymptomatic cases. (C) Histogram of simulated serosurvey data when all infections in a population are severe. (D) Histogram of simulated serosurvey data when one third of all cases is asymptomatic and two thirds severe.

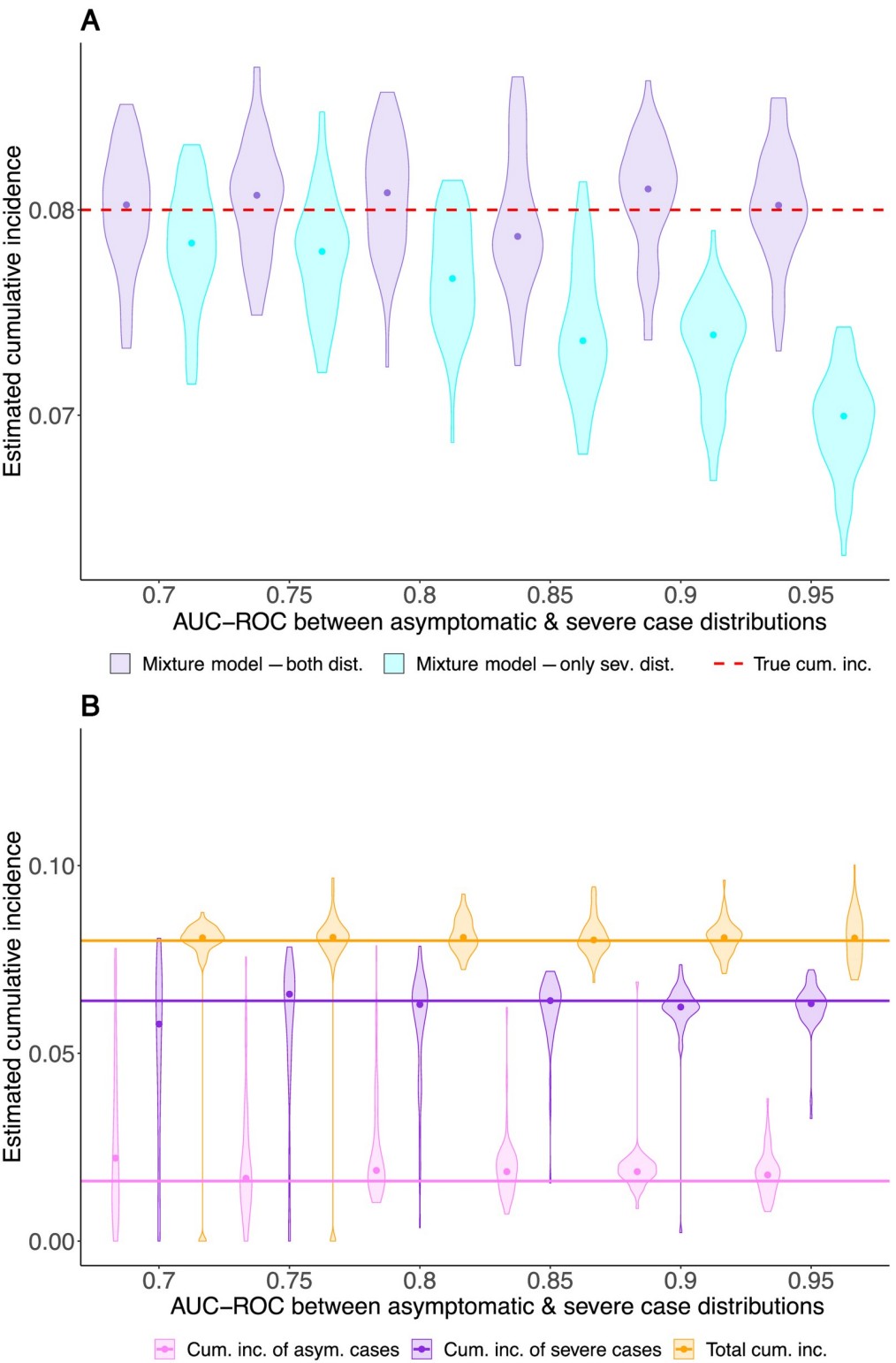

**Fig 6. Estimates of the cumulative incidence in a population where individuals have been uninfected, as well as symptomatically and severely infected.** The x-axes represent the AUC-ROC value between the asymptomatic and severe case distribution. The AUC-ROC value between the control and the severe case distributions is 1. Each violin represents the result of 50 simulated serosurveys with 10, 000 individuals per serosurvey. The true total cumulative incidence of severe and asymptomatic infections is 10%, of which 20% are asymptomatic. (A) Cyan violins show

estimates of the total cumulative incidence based on an inferred case distribution containing only severe case sera, whereas purple violins show estimates where the case distribution is containing both asymptomatic and severe case sera. (B) The estimated cumulative incidence of the mild (light purple) and the severe (dark purple) cases, where the case sera distribution is only based on severe cases, but the likelihood equation also estimates the shape of the asympotomatic cases and their relative prevalence.

used Rogan-Gladen correction and the Bayesian framework that accommodates the sensitivity and specificity alleviate the biases, the estimates of the cumulative incidence have generally wider confidence or high probability intervals than the mixture-model estimates. This is especially pronounced for low levels of cumulative incidence and low test accuracies, and should therefore be of value specifically for SARS-CoV-2 serosurveys early in the pandemic.

Paired with the simulation of serosurveys, the mixture model can provide guidance on the design of serosurveys. It reveals how the test's accuracy, the number of individuals in the serosurvey, and the number of control and case sera affect in the statistical power of the serosurvey. First, we found that a low test accuracy can, up to a level, be compensated by higher sample sizes in the serosurvey. Specifically, a compensation is possible for AUC-ROC values higher than 0.9, but even for these relatively accurate tests the increase in required sample size can be large. Thus, serosurveys need to employ tests of sufficiently high accuracy to be feasible, and increasing a tests accuracy pays off in terms of smaller serosurveys. Secondly, we showed how the number of control and case sera used in the validation of the test influence the statistical power of a serosurvey. Please note that the increasing the number of case and control sera in test validation do not make the test more accurate, but provide more exact estimates of the test's accuracy. For example, to obtain a statistical power of 0.9 for a serosurvey performed at a true cumulative incidence level of 8% and a test accuracy of AUC-ROC = 0.975, 3, 000 individuals need to be enrolled if the test has been validated on 750 control and case serum samples. If the number of control and case sera used for test validation is equal to 2, 500 each, only 1, 000 individuals would have to be enrolled in the serosurvey for the same statistical power. It is therefore worthwhile to carefully weigh the benefits of more thorough test validation against that of expanding the serosurvey. The fact that the number of serum samples used to determine the accuracy of a serological test affects the feasibility and statistical power of serosurveys is an argument for a collaborative scientific effort to establish large, standardized, open validation data sets for SARS-CoV-2 serological tests.

The mixture-model approach as well as the cutoff-based methods rely on representative data from cases. Representative means this cohort should contain individuals who have undergone severe, mild and asymptomatic infections, and the proportions of the infections with different severities should recapitulate the proportions in the population. We have illustrated that an over-representation of severe cases compared to asymptomatic cases in the validation of the serological test leads to underestimation of the cumulative incidence. Thus, it is preferable to include cases detected by contact tracing rather than by more biased detection channels, such as hospitalisation.

Unlike the cutoff-based methods, however, the mixture-model approach allows us to infer the presence of cases that were not used for the validation of the test. To this end, the likelihood of the mixture model is extended by an unknown distribution of test measures. In addition to being able to identify a discrepancy of test validation versus serosurvey data, this extended likelihood also allows for the estimation of the cumulative incidence of the subpopulation with the unknown distribution (Fig 6B). This is crucial for a reliable estimation of the true extent of the current SARS-CoV-2 pandemic. The mixture model can be easily extended to include more than two types of cases. We recommend to always use a mixture model with an additional unknown distribution and to test its fit with a likelihood-ratio test against a null model that

includes all known case distributions. This guards against strong biases arising from neglecting a large fraction with distinct test measure distributions. However, the biological interpretation of an additional distribution is complex. In addition to the severity of infection, many alternative factors could influence the antibody response levels. Most notably, antibody levels have been found to drop over time [29] which can introduce heterogeneities in test measure distributions over time.

The mixture-model approach is applicable over a broad range of serological tests and accepts any common type of quantitative test measures of specific antibody levels, such as optical densities or "arbitrary units" in ELISAs or neutralization titers that inhibit viral replication to any degree specified (e.g NT50 or NT90) in neutralization assays. It is not essential that these measures are linearly correlated with the level of antibodies. It is important, however, that the isotype, units and scales are the same as those for the test measure distributions of control and case sera. While the mixture-model approach has been developed to increase the reliability of cumulative incidence estimates especially for tests with low accuracy, it of course also works for perfect tests.

Similar to the cutoff-based methods, it has been shown that the mixture model can be easily extended by categorical or continuous covariates of the cumulative incidence, such as sex or age [17–23]. Also temporal changes in cumulative incidence—as predicted, for example, by epidemiological models—can be incorporated into this framework. The mixture model provides a more natural approach to integrating multiple test measures, such as IgA and IgG levels. In cutoff-based methods, in contrast, including multiple measures may require complex cutoff functions that depend on these multiple measures and will complicate the determination of sensitivity and specificity.

This work as well as other similar studies that investigate the effect of test accuracy on estimating cumulative incidence highlight the necessity of integrating the development of serological tests with the design of the serosurveys, in which they will be applied [30]. This is especially relevant during the current SARS-CoV-2 pandemic as the serological tests are developed at the same time as the serosurveys are being conducted.

## Methods

### The likelihood of the mixture model

The mixture model approach to inference relies directly on the quantitative measures obtained from serological tests [18–21]. It estimates the cumulative incidence ($\pi$) by maximizing the likelihood of observing the quantitative test measures in the serosurvey data, given the distribution of known case and control sera (see Fig 7A). The likelihood for the data is shown in Eq 1.

$$ll(U) = \sum_{i=1}^{i=n} \log \left( p(\sigma_i = 1|\pi) \, p(U_i|\sigma_i = 1) + p(\sigma_i = 0|\pi) \, p(U_i|\sigma_i = 0) \right) \qquad (1)$$

Here, $U$ is a vector with the quantitative test measures of all $n$ tests, $\sigma$ is a binary vector of length $n$ with their underlying true serological status (1 for infected and 0 for not infected). The probabilities $p(U_i|\sigma_i = 1)$ and $p(U_i|\sigma_i = 0)$ capture the distributions of quantitative test measures of control and case sera, and $p(\sigma_i = 1|\pi)$ and $p(\sigma_i = 0|\pi)$ denote the probability of sampling individuals who have been truly infected or not. The distributions of the quantitative test measures of control and case sera are derived from the observations of known cases and controls. The units of $U$ can be anything that is commonly used in serological tests, such

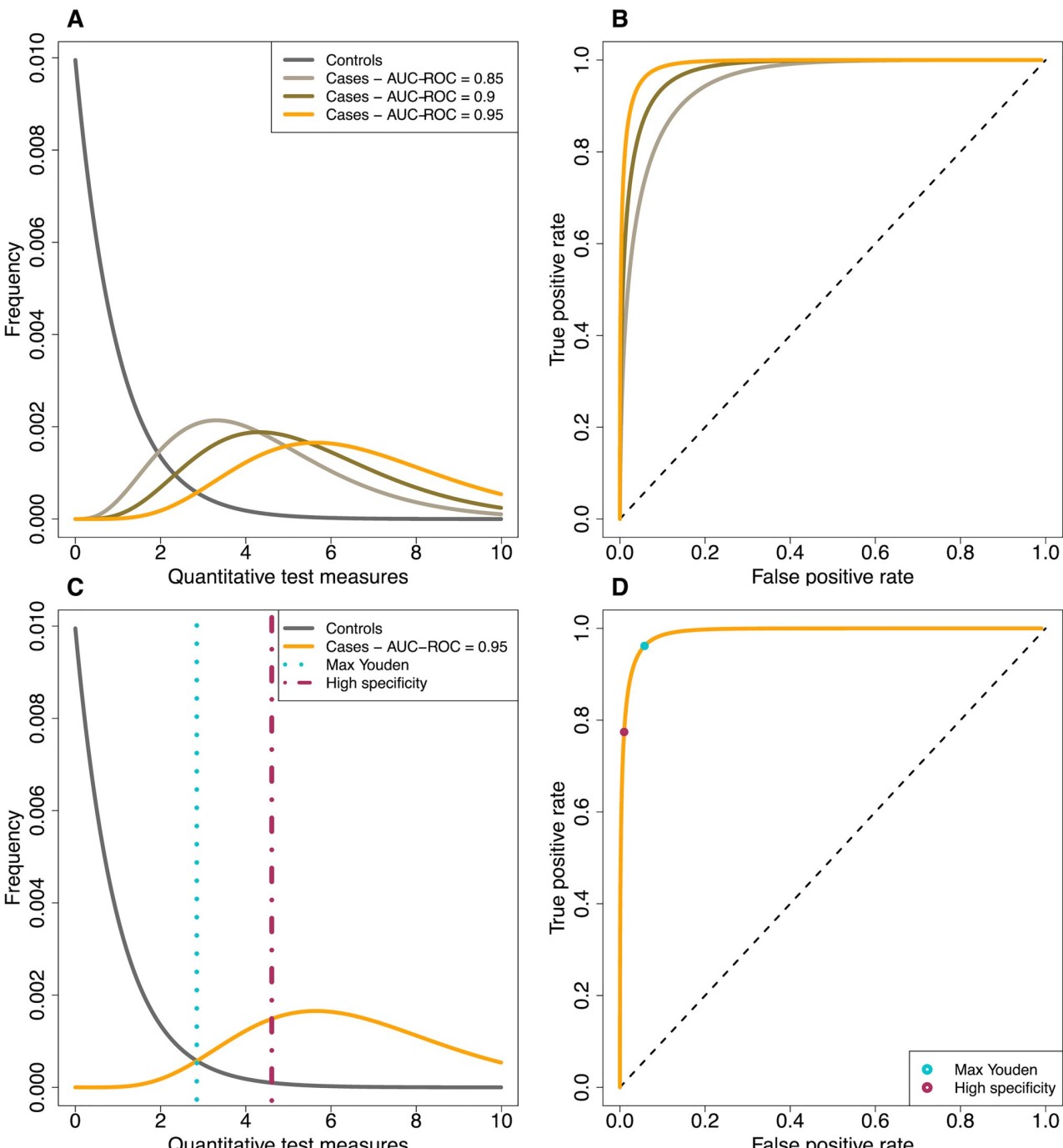

**Fig 7. Conceptual diagram of the distribution of the quantitative test measures for control and case sera.** (A) Hypothetical probability density of quantitative test measures of control sera and three possible case sera distributions. (B) ROC-curves corresponding to the distribution of quantitative test measures of the control sera and each of the possible distributions for the case sera. (C) Visualization of the 'maximal Youden' and 'high specificity' cutoffs. (D) Visualization of the 'maximal Youden' and 'high specificity' cutoffs in the ROC curves.

as optical densities obtained from ELISAs or neutralization titers obtained in neutralization assays.

## Cutoff-based methods

For the cutoff-based methods, the quantitative test measures are dichotomized into seropositives and negatives using a cutoff. There are many ways to estimate a cutoff value for a test [31, 32]. One strategy is to set the cutoff such that the test is highly specific (99%) (see Fig 7C). This is equivalent to minimizing the number of false positives. This will go at the cost of the sensitivity and will lead to more false negatives. This method is referred to as the 'high specificity' method throughout this study.

Another method that is often used to determine the cutoff is to maximize the Youden index (Youden index = sensitivity + specificity − 1 [33]) (see Fig 7C). Graphically, this is equivalent to maximizing the distance between the diagonal and the receiver-operator characteristic (ROC) curve (see Fig 7D). This method is referred to as the 'max Youden' method throughout this study.

To estimate the cumulative incidence in the data with a cut-off based method, the binomial cumulative incidence estimate $q$ is corrected for the sensitivity and specificity of the test. The standard correction is described by Rogan and Gladen [8]:

$$\pi = \frac{q + s - 1}{r + s - 1}.$$

(2)

Here, $r$ is the sensitivity and $s$ the specificity of the test, and $\pi$ is the corrected estimator of cumulative incidence. Effectively, this correction adds the expected number of false negative and subtracts the expected number of false positives from the number of observed positives. The correction works best when the expected number of false positives is smaller than the observed number of positives. We apply this correction to the maximal Youden and the high specificity method and refer to it as the ad-hoc correction.

An alternative way to account for the sensitivity and specificity of the serological test is by using a Bayesian approach which simultaneously estimates the cumulative incidence and the sensitivity and specificity of the serological test [12–14, 16, 30]. This is done based on the sero-survey data and the test validation data. Eq 3 provides the joint posterior distribution of these three parameters, as given in Larremore et al. [30]. Here, $X$ represents the serosurvey data, $V$ the validation data, $n^+$ the number of positive tests in the serosurvey data, $N_{fields}$ the number of individuals in the serosurvey, $N_{neg}$ the number of control sera, $N_{pos}$ the number of case sera, $tp$ the number of true positives in the cases sera and $tn$ the number of true negatives in the control sera. We apply this method using the cut-offs calculated with the maximal Youden and the high specificity method.

$$Pr(\pi, s, r | X, V) \propto \left( (1 - s + \pi(r + s - 1))^{n^+} (s - \pi(r + s - 1))^{N_{fields} - n^+} \right) *$$

$$s^{tn}(1 - s)^{N_{neg} - tn} r^{tp} (1 - r)^{N_{pos} - tp}.$$

(3)

## Simulations

We simulate serosurveys by assuming given cumulative incidences and distributions of the quantitative test measures for control and case sera. For each virtual individual enrolled in these *in silico* serosurveys, we conduct a Bernoulli trial with the probability of success set to the chosen cumulative incidence. In a second step, we simulate the serological test of each

individual. To this end, we draw a random quantitative test measure from the distribution of either case or control sera—depending on whether the individual is truly seropositive or not.

Besides the serosurvey, we also simulate data of known control and case sera. Unless otherwise specified we simulated 5, 000 control as well as case sera. For each of these 5, 000 control and case sera we randomly drew a value from the assumed distribution. We assume the distribution of quantitative test measures for the control sera to be $\Gamma$-distributed with a shape and scale parameter of 1, which results in a mean of 1, and the distribution for case sera to be $\Gamma$-distributed with varying shape parameters and a scale parameter of 1. The scale parameter of the distribution for case sera determines the amount of overlap between the quantitative test measure of cases with controls, and thus modifies test accuracy as measured by AUC-ROC (Fig 7).

The simulated validation data are used to perform the estimation of the cumulative incidence with the methods described above. For the mixture model, we fitted a density distribution through the control and case sera data with the function 'density' in R [34]. For the Rogen-Gladen correction, we use the simulated control and case sera to calculate the sensitivity and specificity of the test. For the Bayesian method, we use the simulated control and case sera to include also the uncertainty in the sensitivity and specificity of the test in the cumulative incidence estimate.

## Confidence interval

We suggest a two-step bootstrap procedure to calculate a confidence interval for the estimated cumulative incidence inferred with the mixture model and the ad-hoc corrected cutoff-based methods. First, we re-sample the test measures of control and case sera, and use the re-sampled dataset to obtain a new fitted density distribution for both the control and case sera. Second, we re-sample the test measures of the serosurvey, and, using the new density distributions of the test measures for the control and case sera, obtain bootstrap estimates of the cumulative incidence. This procedure makes sure that the uncertainties arising from the lack of test accuracy as well as the lack of precision in the test measure distributions are appropriately accounted for in the estimation of the cumulative incidence. An R-function that performs this procedure is included in the supplementary R-package.

In the results, we compare the mixture model to a Bayesian implementation of the cutoff based methods that includes uncertainty in the sensitivity and specificity [15, 16]. These methods provide credible intervals instead of confidence intervals.

## Power analysis

To estimate the statistical power of a serosurvey, we conducted 3, 000 serosurveys for each given test accuracy and sample size. The power is determined as the proportion of simulations for which the cumulative incidence is estimated successfully. We define an estimate to be successful when it deviates less than 25% from the true cumulative incidence and the true cumulative incidence is within 2 standard deviations of the estimate.

## Incorporating distinct distributions for asymptomatic and severe case sera

Eq 4 represents a likelihood that allows for the possibility that there are various types of case sera with distinguishable distributions (See Fig 5B) [23]. For the sake of simplicity, we assume that there are two types of case sera: those from individuals who underwent a severe infection (severe cases) and those from individuals who underwent an asymptomatic infection

(asymptomatic cases).

$$ll(U) = \sum_{i=1}^{i=n} \log \Big( p(\sigma_i = 2 | \pi_S, \pi_A) \, p(U_i | \sigma_i = 2) + p(\sigma_i = 1 | \pi_A) \, p(U_i | \sigma_i = 1) +$$

$$p(\sigma_i = 0 | \pi_S, \pi_A)) \, p(U_i | \sigma_i = 0) \Big) \tag{4}$$

Compared to Eq 1, $\sigma_i$ is now a categorical value that represents whether an individuals is sero-negative (0), seropositive due to an asymptomatic infection (1) or seropositive due to a severe infection (2). $\pi_A$ is the cumulative incidence of asymptomatic infections and $\pi_S$ the cumulative incidence of severe infections. This likelihood is formally equivalent to the one presented in van Boven et al [23], without the age-structure. We used the extended likelihood to estimate the total cumulative incidence, the cumulative incidence of asymptomatic cases and the shape parameter of the distribution of quantitative test measures of the asymptomatic case sera.

The identifiability and confidence of the estimates of these three parameters depends on the amount of overlap between the distribution of the asymptomatic and severe case sera and the overlap between those and the control sera. In our simulations, we hold the area under the ROC-curve between the control and severe case sera constant, at a level of 1, while we vary the area under the ROC curve between the asymptomatic and severe case sera. We set the mean of the distribution of asymptomatic cases always to a lower value than that of the severe cases (see Fig 5B). Both these means are larger than 1, which is the mean of the distribution of test measures for control sera. We use a scale parameter of 1 for both distributions.

## Implementation

The likelihood function for estimating cumulative incidence as well as the simulations were implemented in the R language for statistical computing [34]. The Bayesian correction is implemented with the 'Rstan' package in R [35, 36]. An R-package containing the code can be found here: https://gitlab.ethz.ch/jbouman/pist.

## Supporting information

**S1 Fig. Sensitivity and specificity corresponding to the cutoff-based methods across the range of AUC-ROC values we consider in this study.**
(TIF)

**S2 Fig. Performance of the uncorrected cutoff-based methods.** (A) Estimated cumulative incidence for three levels of true cumulative incidence. (B) Half of the 95% confidence intervals estimated based on the bootstrap method for both uncorrected cutoff-based methods.
(TIF)

## Acknowledgments

We are thankful to Claudia Igler, Christian Althaus, Sarah Kadelka and Michiel van Boven for discussions and comments on the manuscript.

## Author Contributions

**Conceptualization:** Judith A. Bouman, Sebastian Bonhoeffer, Roland R. Regoes.

**Formal analysis:** Judith A. Bouman, Roland R. Regoes.

**Funding acquisition:** Roland R. Regoes.

**Methodology:** Judith A. Bouman, Roland R. Regoes.

**Software:** Judith A. Bouman, Julien Riou.

**Writing – original draft:** Judith A. Bouman, Roland R. Regoes.

**Writing – review & editing:** Judith A. Bouman, Julien Riou, Sebastian Bonhoeffer, Roland R. Regoes.

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
