## [Decision Letter · Decision Letter 0]

9 Jul 2020

Dear Mrs. Bouman,

Thank you very much for submitting your manuscript "Estimating seroprevalence with imperfect serological tests: exploiting cutoff-free approaches" for consideration at PLOS Computational Biology.

As with all papers reviewed by the journal, your manuscript was reviewed by members of the editorial board and by several independent reviewers. In light of the reviews (below this email), we would like to invite the resubmission of a significantly-revised version that takes into account the reviewers' comments.  

We cannot make any decision about publication until we have seen the revised manuscript and your response to the reviewers' comments. In particular, Reviewer 2 raises a number of substantive methodological issues which will need to be addressed to their satisfaction.  Reviewer 1 raises a question of novelty, but perhaps a full response to Reviewer 2's concerns would address this issue. 

Sincerely,

James Lloyd-Smith

Associate Editor

PLOS Computational Biology

Virginia Pitzer

Deputy Editor

PLOS Computational Biology

Reviewer's Responses to Questions

**Comments to the Authors:**

Reviewer #1: The manuscript by Bouman and colleagues, entitled "Estimating seroprevalence with imperfect serological tests: exploiting cutoff-free approaches", presents an investigation of the performance of methods for estimating seroprevalence when data/assays are imperfect.

The main focus of the study is on the comparison of methods based on binary data (being antibody positive or not) and those that use quantitative antibody level data. They use a range of relevant simulations to assess how well different methods work, and show that a likelihood-based mixture model performs particularly well, especially when test specificity and sensitivity are lower.

The study provides an excellent reference on seroprevalence estimation methods, investigates highly relevant aspects in a careful way, is well written and easy to follow. The authors also helpfully provided R functions that help implement the methods (although I wasn't able to try the functions because I couldn't access them in the online review system).

I don't have any major comments on how the study was conducted or presented.

The only reservation I might have is on the novelty aspect of the work.

The main focus of the study is the use of quantitative vs binary antibody data for estimating seroprevalence, and the application of a likelihood-based mixture model approach.

This approach is already well-established as being particularly suitable in a context of imperfect data, and has been used frequently.

The authors clearly acknowledge this and cite relevant work.

The addition of a third "unknown" distribution within the data is a nice addition with a clear application, even if it is not a ground-breaking development.

So it seems that the real novelty, and purpose of this study, is in the formal comparison of the different methods, and in combining it with a power analysis and useful R functions.

When first reading the abstract and main text, I was expecting a fully new approach to this problem, and it took me a while before I realized this is not the purpose of the study. A suggestion to avoid this would be to rephrase it slightly in the abstract and introduction as a review of existing methods, and that the purpose of the study is rather to be a useful reference, as opposed to a new development in this field.

Minor comments:

- According to nomenclature guidelines coronavirus should not be capitalized.

- The results in figure 1 are useful for comparison, but I wonder whether it would be useful to add more formal measures of the errors. Now I am for example left wondering whether or not the likelihood method outperforms the corrected Max Youden method.

- It might be helpful to provide more information in the figure legends so they can be understood more easily.

- Axis tick label fonts are quite small (especially in fig 2), consider increasing.

- I was a bit visually distracted by the grid lines in the figures, but that's probably a highly subjective preference.

- Throughout the text the number 10,000 was written with an inverted comma (10'000), is this a mistake or on purpose?

- I found a few typos but they are tricky to point out due to the lack of line numbers (suggestion: add line numbers whenever sending a document for review).

Reviewer #2: Bouman et al have submitted a manuscript called "Estimating seroprevalence with imperfect serological tests: exploiting cutoff-free approaches" for consideration in PLOS Computational Biology. The manuscript is stimulating at present but is not currently publishable in PLOS Computational Biology, due to issues that are discussed below.

Overall, the argument of the manuscript is that the estimation of seroprevalence can be improved via the use of a cutoff-free / likelihood based approach. The assumption is that the observed scalar data come from a mixture of two or more classes of signals (seropositive and seronegative, in the simplest case), and that each individual data point belongs to one of the classes. In other words, the authors use a one-dimension mixture model with K=2 (negative, positive) or K=3 (negative, mild positive, severe positive). In this context, to determine seroprevalence, one need only infer the mixture weight(s).

Broad Comments:

This manuscript's proposition rests on the establishment of two key claims:

1. The likelihood-based approach works.

2. The likelihood-based approach provides superior estimates.

The manuscript at present has not quite established those claims.

1. That the likelihood-based approach works is not quite established in this paper due to the fact that it assumes (I think?) that continuous responses come from a known distribution. Here, for instance, it appears that OD values are gamma distributed in the simulations. What if gamma are used for inference, but the true values come from a different distribution? What if neither distribution is gamma? Is model selection to learn the class of data-generating distributions possible with sufficient data? What if, e.g., the true distribution of test measures for positive cases is a truncated normal distribution and a gamma distribution is selected for the model?

At present the manuscript notes that one can test to see whether K=2 or more distributions are present in the data, which the authors note is a form of misspecification testing. However, this misspecification argument applies only to the mixture cardinality. A key question is whether the distributions themselves can be misspecified without a substantial loss of performance.

As an example, the distributions of FACS-derived responses from a luminex/multiplex platform are not gamma distributed. Another example comes from the Santa Clara / Stanford SARS-CoV-2 saga (Bendavid et al), which showed (without intending to) that serological responses for case control samples seem to vary by laboratory. In bother of these cases, modeling with a single gamma distribution would be a misspecification.

Does the theory of mixture models rescue this point? If not, the authors need to make the case that their work will still be valuable. If so, these kinds of broader results from mixture model theory should be introduced and discussed.

Put more succinctly, the likelihood method does not assume a distribution for the quantitative test measure given infection status ($U|\\sigma=0$), yet it seems that a gamma distribution is assumed for all simulations. If the data are simulated from a gamma distribution and then a gamma distribution is used in the model fitting, then the current set of results demonstrate what happens when the model matches the data generating process.

Finally, if one were to find a mixture of 3 classes, how would one know what the different classes correspond to, if there are only 2 classes in the training data? For the purposes of calculating a seroprevalence estimate, for instance, it would be important to know, a priori, whether some of the middling results were due to cross-reactivity with other (e.g. seasonal) coronavirues, due to loss of IgG reactivity, or due to mild infection. In other words, the class labels are important. Perhaps the authors meant that the presence of a third group, not represented in the controls, could be tested for, but that interpretation would be difficult to establish?

2. That the likelihood-based approach provides superior estimates is not quite established either. The paper focuses on showing that the mixture model provides typically more reliable inferences, i.e. less variable estimates of seroprevalence, and trends. How do the Bayesian estimation procedures compare? Bayesian methods are referred to in the final supplemental figure, but comparisons are not found in the main text.

The main argument for using the likelihood-based approach (graphically) appears to be found in Figures 1 and 2, based on the distributions of estimates and their biases and variances. The Bayesian methods (which incorporate a likelihood function that includes a cutoff, but does not assume a generating distribution for the data) should be included for comparison.

More broadly, on the point of comparisons, it is widely understood that using "raw" prevalence estimates is a bad (and biased) idea. We suggest removing all results for the uncorrected methods. This will simplify the message, and avoid the inclusion of a "straw man" estimates. Fig 1, for instance, would be much more readable without the Max-Youden estimates.

As an aside, the Bayesian methods that the authors use appear to be estimation procedures, rather than corrections.

Reproducibility:

- The R code cannot be found online or in the manuscript's materials.

Discussion or broader questions:

- Recent analysis of SARS-CoV-2 serological data suggest that IgG responses wane over time. No changes are needed to the manuscript, per se, but this point may be valuable to discuss in future submissions, particularly given the K=3 mixtures.

- Issues with learning distributions for positives from only the symptomatic/confirmed cases sounds like an overfitting problem--the lessons from the case control training data do not generalize to the population, in other words. This language may or may not be valuable for bringing in intuition for the ML types out there.

- Paragraph 3 of introduction could be updated to reflect some of the numerous recent developments.

Other:

- Page 8 mentions one-step Bayesian model incorporating uncertainty from test validation into prevalence estimates, but no citations are provided. These references appear to do exactly that.

- Gelman, Carpenter. Bayesian analysis of tests with unknown specificity and sensitivity. medRxiv, 2020.

- Larremore, Fosdick, Zhang, Grad. Jointly modeling prevalence, sensitivity and specificity for optimal sample allocation. bioRxiv, 2020.

- Any analysis of real data would do a lot to strengthen the claims of the authors here.

- Figures could use clarification. Font sizes were small (Fig 2), and line widths or dot sizes made some plots very hard to read (Fig 2), relative to grid lines, for instance.

- e.g. weird fonts Fig 6A y axis?

- Fig 4B... are the x-axis values 75, 100, 150, but presented for each color? Or do the values interpolate between 75, 100, 150, and so on? My guess is the former, but this is a little unclear.

- What are the subplots of Fig1? No notes in caption.

- Various formatting issues in the references, FYI. For instance {SARS-CoV-2} is often uncapitalized.

**Have all data underlying the figures and results presented in the manuscript been provided?**

Reviewer #1: None

Reviewer #2: Yes

PLOS authors have the option to publish the peer review history of their article (what does this mean?). If published, this will include your full peer review and any attached files.

Reviewer #1: **Yes: **Benny Borremans

Reviewer #2: No
---

## [Decision Letter · Decision Letter 1]

24 Nov 2020

Dear Mrs. Bouman,

Thank you very much for submitting your manuscript "Estimating cumulative incidence of SARS-CoV-2 with imperfect serological tests: exploiting cutoff-free approaches" for consideration at PLOS Computational Biology. As with all papers reviewed by the journal, your manuscript was reviewed by members of the editorial board and by several independent reviewers. The reviewers appreciated the attention to an important topic.

Based on the reviews, we plan to accept this manuscript for publication; however, we first ask that you attend to the final comments generously provided by the reviewers. Most of them seem very straight-forward to address, and will enhance the clarity of the published paper.  You will note that one reviewer is skeptical about the new title and framing in terms of cumulative incidence; I understand and support the argument that you put forward in the cover letter for your resubmission, but it seems they did not get this perspective from the manuscript itself.  You might consider adding a sentence or two of discussion about the distinction between cumulative incidence and seroprevalence, and how your model relates to each of them.

Sincerely,

James Lloyd-Smith

Associate Editor

PLOS Computational Biology

Virginia Pitzer

Deputy Editor

PLOS Computational Biology

[LINK]

Reviewer's Responses to Questions

**Comments to the Authors:**

Reviewer #1: The revised version of the manuscript includes some great improvements, and I only have two minor suggestions.

Page 2 paragraph 3: It might be confusing to refer to a single specific Bayesian application as "the Bayesian framework", given that Bayesian frameworks can be applied to any system or situation. I understand that this is how the authors choose to refer to it within the context of the study, but this is not obvious from the text. A simple change to "we here refer to as .." would probably help to make this clear. Same thing for mixture model in the following paragraph.

Table S1 is a great addition, I suggest moving it to the main text if there is space.

Reviewer #2: Bouman et al have submitted a revision to their manuscript called "Estimating seroprevalence with imperfect serological tests: exploiting cutoff-free approaches." The manuscript is interesting, and their revision makes improvements upon the first submission. Actually, to call this just a "revision" would be to undersell the big changes that the authors have made across the paper and code. In short, I think the manuscript is a lot stronger, ties in nicely with existing literature, but makes its own contribution clear. This work is likely to be of interest to those who are working with SARS2 ELISA (and other) serological data, and I therefore recommend publication.

In particular, I want to thank the authors for moving beyond the gamma distributions that were used in the first manuscript to try to be more general. I can see that this was a lot of work, and appreciate the effort.

The manuscript could be accepted as is, but I also have a few suggestions, which are minor:

1. In the introduction, "the post-hoc Rogan-Gladen correction" is referred to as if readers will know it by name. A slightly more gentle introduction to what the correction is, or is meant to do, might be nice, as context.

2. Page 3 "specificty" typo.

3. Page 5 "a certain statistical power of (...) decreases" This sentence doesn't read well without the (parenthetical).

4. Given the rapid acceleration of cases across much of the northern hemisphere, the end of paragraph 1 in discussion should suggest that this will be of value specifically for *early pandemic* SARS-CoV-2 serosurveys, since seroprevalence is likely to be higher.

5. Finally, I wasn't quite sure that I understood the discussion about the inference of cases that were not included in the validation of the test. Methodologically, how should one do this? (e.g. AIC or BIC or some other model selection / complexity control?) I think the point is made in a nice visual way in Fig 5, but it might be worth at least discussing ways that people could look to see whether their dataset supports a third distribution. By the way, should the title on Fig 5B be slightly different to note that there are asymptomatic controls too?

As a final note, I noticed that the title is now about cumulative incidence instead of seroprevalence, but given that there is evidence of (typical) waning antibody titers over time, from some studies, I wonder if perhaps seroprevalence is a more appropriate word than cumulative incidence. I fear that cumulative incidence may not be quite as accessible via serology after nearly a year, as we may have initially hoped. I leave it to the authors and editor to decide, of course.

In sum, despite a few minor suggestions, the work by Bouman et al is a valuable contribution to the serosurvey literature. Its blend of practical application, theoretical development, and a commitment to open-source tools make it a great fit for PLOS Computational Biology. I recommend publication.

**Have all data underlying the figures and results presented in the manuscript been provided?**

Reviewer #1: Yes

Reviewer #2: Yes

PLOS authors have the option to publish the peer review history of their article (what does this mean?). If published, this will include your full peer review and any attached files.

Reviewer #1: No

Reviewer #2: No
---

## [Editor Report · Decision Letter 2]

20 Jan 2021

Dear Mrs. Bouman,

We are pleased to inform you that your manuscript 'Estimating the cumulative incidence of SARS-CoV-2 with imperfect serological tests: exploiting cutoff-free approaches' has been provisionally accepted for publication in PLOS Computational Biology.  Thanks for your prompt and on-target responses to the reviewer and editor comments.

Before your manuscript can be formally accepted you will need to complete some formatting changes, which you will receive in a follow up email. A member of our team will be in touch with a set of requests.  As you do so, please correct a typo in the next text you've added at the start of the discussion ('harbor' is misspelled as 'habor').

Best regards,

James Lloyd-Smith

Associate Editor

PLOS Computational Biology

Virginia Pitzer

Deputy Editor-in-Chief

PLOS Computational Biology

---

## [Editor Report · Acceptance letter]

22 Feb 2021

PCOMPBIOL-D-20-00844R2 

Estimating the cumulative incidence of SARS-CoV-2 with imperfect serological tests: exploiting cutoff-free approaches

Dear Dr Bouman,

I am pleased to inform you that your manuscript has been formally accepted for publication in PLOS Computational Biology. Your manuscript is now with our production department and you will be notified of the publication date in due course.

With kind regards,

Alice Ellingham
